# Development of SNP Markers for White Immature Fruit Skin Color in Cucumber (*Cucumis sativus* L.) Using QTL-seq and Marker Analyses

**DOI:** 10.3390/plants10112341

**Published:** 2021-10-29

**Authors:** D. S. Kishor, Hemasundar Alavilli, Sang-Choon Lee, Jeong-Gu Kim, Kihwan Song

**Affiliations:** 1Department of Bioresources Engineering, Sejong University, Seoul 05006, Korea; kishoreflmes@gmail.com (D.S.K.); alavilli.sundar@gmail.com (H.A.); 2PHYZEN, Seongnam-si 13558, Gyeonggi-do, Korea; sclee0923@phyzen.com; 3National Academy of Agricultural Science, Rural Development Administration, Jeonju 54874, Korea; jkim5aug@korea.kr

**Keywords:** cucumber, QTL-seq, SNP markers, white immature fruit skin color

## Abstract

Despite various efforts in identifying the genes governing the white immature fruit skin color in cucumber, the genetic basis of the white immature fruit skin color is not well known. In the present study, genetic analysis showed that a recessive gene confers the white immature fruit skin-color phenotype over the light-green color of a Korean slicer cucumber. High-throughput QTL-seq combined with bulked segregation analysis of two pools with the extreme phenotypes (white and light-green fruit skin color) in an F_2_ population identified two significant genomic regions harboring QTLs for white fruit skin color within the genomic region between 34.1 and 41.67 Mb on chromosome 3, and the genomic region between 12.2 and 12.7 Mb on chromosome 5. Further, nonsynonymous SNPs were identified with a significance of *p* < 0.05 within the QTL regions, resulting in eight homozygous variants within the QTL region on chromosome 3. SNP marker analysis uncovered the novel missense mutations in *Chr3CG52930* and *Chr3CG53640* genes and showed consistent results with the phenotype of light-green and white fruit skin-colored F_2_ plants. These two genes were located 0.5 Mb apart on chromosome 3, which are considered strong candidate genes. Altogether, this study laid a solid foundation for understanding the genetic basis and marker-assisted breeding of immature fruit skin color in cucumber.

## 1. Introduction

Cucumber (*Cucumis sativus* L.; 2*n* = 2*x* = 14) is a major economically important vegetable crop in the *Cucurbitaceae* family. Its total annual global production is 87,805,086 tons in an area of 2231,402 ha, and 80 % of global production comes from China with a yearly output of 70,338,971 tons (FAO, 2019). Cucumber fruits are usually consumed fresh or as processed pickles after 8–18 days of anthesis [1].

Cucumber fruits display wide phenotypic variation in immature fruit skin color from dark green to white appearance [2]. The green skin color of the cucumber fruit is directly associated with the accumulation of chlorophyll [1,3,4,5]. The external skin color of immature fruit is considered as an essential quality trait that decides consumer preference. In order to develop cucumber varieties with various skin colors, it is necessary to understand the inheritance and genes governing the fruit skin color.

Upon the availability of the genomic information of cucumber [6,7,8], several genes were identified, which led to the development of molecular markers for marker-assisted selection (MAS) of fruit skin color in cucumber. A single dominant gene *B* (R2R3-MYB) was identified for the orange mature fruit skin color on chromosome 4, and an Insertion-deletion (InDel) marker was developed based on the 1-bp deletion in the third exon of this gene, was co-segregated with fruit skin color [9]. High-resolution mapping for the dull fruit skin trait in cucumber revealed a single dominant gene *D* on chromosome 5 between markers SSR37 and SSR112, at a physical distance of 244.9 kb [10]. The *D* gene was reported to be closely linked to the genes governing fruit wart (*Tu*), uniform immature fruit color (*u*), and small spines (*ss*) [11,12].

Studies on green fruit skin trait have shown that SNPs in *ARC5* and *Ycf54* genes cause light-green immature fruit skin color in cucumber [3,13]. Yang et al. [14] studied the inheritance of the uniform immature fruit color in cucumber and identified a single recessive gene that confers uniform immature fruit color (*u*) phenotype. *u* gene was mapped to a 313.2 kb on chromosome 5 between co-dominant SSR markers SSR10 and SSR27 at a genetic distance of 0.8 and 0.5 cM, respectively [14], which further laid a strong foundation for marker-assisted breeding of uniform immature fruit color trait in cucumber.

Recent studies have shown that a single gene controls the external fruit color trait and white skin color is recessive over dark green skin color in cucumber [2,15,16]. A frameshift mutation, which leads to a premature stop codon in the *w* gene (*aprr2*) on chromosome 3, was reported to be a sole candidate gene responsible for white immature fruit color phenotype in cucumber and associated with chlorophyll biosynthesis [15]. Tang et al. [16] studied the genetics of white immature fruit color in cucumber and mapped a single recessive gene (*w_0_*) for white immature fruit color on chromosome 3 to approximately 100.3 kb between two flanking markers, Q138 and Q193 [16]. In spite of several studies, Tang et al. [16] proposed that further study is necessary to understand the genetic basis of white immature fruit skin color in cucumber.

This study determined a novel genetic architecture for the white immature fruit skin color of cucumber by using QTL-seq and *SNP marker* analyses, which revealed that white fruit skin color trait is recessive over light-green fruit skin color of Korean slicer cucumber. Furthermore, this study identified a novel allelic variant of the *Chr3CG52930* gene and a new candidate gene, ‘*Chr3CG53640*’, for white fruit skin color trait in cucumber. This finding would facilitate understanding of the genes involved in cucumber skin color and contribute to the development of cucumbers using marker-assisted breeding.

## 2. Materials and Methods

### 2.1. Genomic DNA Extraction and Pooling

Genomic DNAs were extracted from leaves of the two parental lines (Inbred line of Korean slicer cucumber ‘MEJ’ with light-green skin color and ‘PI525075′ with white skin color) and their F_2_ plants using a cetyltrimethylammonium bromide (CTAB) method [17]. The integrity of the extracted genomic DNA was checked by an agarose gel electrophoresis. Genomic DNA (gDNA) was quantified using Quant-iT™ PicoGreen™ dsDNA Assay Kit (Invitrogen, Waltham, MA, USA) and Qubit fluorometer (Thermo Fisher Scientific, Waltham, MA, USA). Genomic DNAs of 16 F_2_ plants with white immature skin color were mixed with equal amounts and used as white skin-pool, and genomic DNAs of 20 F_2_ plants with light-green skin color were mixed with an equal quantity and used as light-green skin-pool for downstream analysis.

### 2.2. Whole-Genome Resequencing

Quality and quantity of gDNAs of two parental lines and pooled DNA samples were examined again using an agarose gel electrophoresis and Qubit fluorometer (Thermo Fisher Scientific, USA). The sequencing libraries were prepared according to the TruSeq DNA PCR-free Sample Preparation Kit (Illumina, San Diego, CA, USA). Briefly, fragmentation of 1µg of genomic DNA was performed using adaptive focused acoustic technology (AFA; Covaris, Woburn, MA, USA), and the fragmented DNA was end-repaired to create 5′-phosphorylated, blunt-ended dsDNA molecules. Following end-repair, DNA was size-selected with the bead-based method. These DNA fragments go through the addition of a single ‘A’ base and ligation of the Truseq indexing adapters. The purified libraries were quantified using quantitative PCR (qPCR) according to the qPCR Quantification Protocol Guide (KAPA Library Quantification kits for Illumina Sequencing platforms) and qualified using the high-sensitivity DNA chip (Agilent Technologies, Santa Clara, CA, USA). The paired-end (2 × 150 bp) sequencing was performed using the HiSeq-X platform (Illumina, San Diego, CA, USA) by the Macrogen Co. (Seoul, Korea).

### 2.3. QTL-seq Analysis

QTL-seq analysis was performed using the QTL-seq program (version 2.1.3, https://github.com/YuSugihara/QTL-seq) (access on 7 October 2021) with default parameters as described previously [18]. Briefly, high-quality sequencing data were obtained by trimming the raw sequencing data using the Trimmomatic program [19] in the QTL-seq program. The high-quality sequencing data of the PI525075 parent with white skin color were aligned to the Korean cucumber genome (*Cucumis sativus* var. JEF, BioProject no: PRJNA732224, https://www.ncbi.nlm.nih.gov/bioproject/732224) (access on 7 October 2021) using BWA [20]. Then variants from PI525075 parent were used to generate the PI525075 reference genome by substituting the variant bases in the Korean cucumber genome. The high-quality sequencing data of two pooled DNA samples (white skin-pool and light-green skin-pool) were mapped to the PI525075 reference genome and variants (SNPs and InDels) were detected. The SNP index at each SNP position was calculated and then Δ (SNP index) was calculated using the formula: [SNP index (white skin-pool)—SNP index (light-green skin-pool)]. The average SNP index and Δ (SNP index) distribution were estimated in a given genomic interval using a sliding window approach with 2 Mb window size and 100 kb increment and plotted to generate SNP index plots for all chromosomes. The candidate genomic regions for the phenotype were determined based on the sliding window plots. The regions in which the average Δ (SNP index) was significantly greater than the surrounding region and exhibited an average *p* < 0.05 were considered as candidate QTLs.

Variants’ information in the candidate QTLs was extracted from the variant calling file (VCF) generated by the QTL-seq program. Then, variants with *p* < 0.05 were further selected and used for further downstream analysis.

### 2.4. Annotation of Variants and Identification of Variants Causing Protein Sequence Change

The annotation of variants was performed using SnpEff software (version 5.0e, [21]). The variants present in gene regions (from 5’ UTR to 3’ UTR, including intron and exon) was annotated as genic, while other genomic regions were intergenic. Variants in coding sequences (CDSs) were further divided into synonymous and non-synonymous.

Variants that caused the change of amino acid in deduced protein sequence of the genes were identified among the selected variants with *p* < 0.05 in the QTL regions under the following criteria: (1) variants that were different between bulk1 (white skin) and bulk2 (light-green skin) were selected; (2) among the selected variants, only homozygous variants were selected; (3) homozygous variants that caused the amino acid change in protein sequences encoded by genes were finally selected based on variant annotation information as candidate variants associated with the phenotype.

### 2.5. Development of Molecular Markers

The flanking sequences of variants were extracted from the JEF Korean cucumber genome sequences and then used to design cleaved amplified polymorphic sequence (CAPS) and derived cleaved amplified polymorphic sequence (dCAPS) primers. CAPS and dCAPS primers were designed using in-house scripts modified from CAPS-finder.pl (https://github.com/mfcovington/CAPS-finder/blob/master/CAPS-finder.pl) (access on 7 October 2021) and dCAPS Finder (http://helix.wustl.edu/dcaps/dcaps.html) (access on 7 October 2021), respectively. Parameters of primer design were a primer size of 17– 25 mer, GC% of 50%, Tm of 50–60 °C, and amplicon size of 200–700 bp. BLASTN searches against reference genome sequences confirmed the specificity of designed primers. Designed molecular markers were validated by agarose gel electrophoresis after genomic DNA PCR and restriction enzyme treatment with genomic DNAs of the parents and F_2_ plants.

## 3. Results

### 3.1. Inheritance of Immature Fruit Skin Color

The phenotype of the two parental lines, MEJ and PI525075, along with their F_1_ (MEJ/ PI525075), were shown in Figure 1a. The skin color of MEJ and PI525075 were evaluated for two skin color indices (one for white and two for light-green) at the immature fruit stage. The fruits of F_1_ derived from a cross between MEJ and PI525075 were demonstrated to have a light-green immature fruit skin. In the F2 population, there were 106 and 30 plants with light-green skin and white skin, respectively, which was fit to a segregation ratio of 3:1. This implies the recessive nature of the gene for white immature fruit skin color (Table 1 and Figure 1b).

### 3.2. Whole-Genome Re-Sequencing and QTL-seq Analysis

For whole-genome re-sequencing, 16 plants with white immature fruit skin and 20 plants with light-green immature fruit skin were selected among the 136 F_2_ plants (Figure 1b) and their gDNAs were then pooled respectively to prepare the white fruit skin-pool and light-green fruit skin-pool along with their two parental lines, MEJ and PI525075. Whole-genome re-sequencing results showed that a total of 192,314,600 and 191,719,994 raw reads were generated for PI525075 and MEJ, respectively, while a total of 204,951,660 and 228,647,910 raw reads were obtained for white fruit skin-pool and light-green fruit skin-pool, respectively. Upon trimming, a total of 157,678,304 reads were generated from PI525075; 187,218,180 reads from MEJ; 175,957,140 reads from white fruit skin-pool, and 197,713,150 reads from light-green fruit skin-pool, each corresponding to more than 20 Gb read length, and more than 75% of the reads were clean reads (Table 2). Clean reads from the parents and bulks were compared to the estimated cucumber genome size of 350 Mb, which indicated that all genomes were sequenced at a depth ranging from 64.32X to 80.44X. In comparison with the PI525075 reference genome using QTL-seq program, we have identified a total of 160,392 SNPs for white fruit skin-pool and 120,899 SNPs for light-green fruit skin-pool across the chromosomes (Table 3).

To identify the genomic region governing immature fruit skin color trait between the white fruit skin-pool and light-green fruit skin-pool, the SNP index of individual SNPs were calculated using the parental PI525075 line as a reference genome and compared these with the bulks’ sequences. An SNP index of zero represents entire short reads of the white fruit skin genome, whereas an SNP index of one indicates that the reads are from the light-green fruit skin genome. The average SNP index was estimated in a given genomic interval with 2 Mb window size and 100 kb increment and plotted to generate SNP index plots for white fruit skin and light-green fruit skin pools against all chromosomes (Appendix A). To identify the differences in the SNP-indices of the two pools, Δ (SNP index) was estimated by combining the SNP index information of white fruit skin and light-green fruit skin pools and a statistical confidence interval was plotted against the reference genome of cucumber (Appendix A). Furthermore, significant genomic regions were detected at a statistical significance of *p* < 0.05 according to the principle of SNP index estimation showed in QTL-seq analysis (Appendix A), resulting in two significant genomic regions harboring candidate QTLs for white fruit skin trait within the genomic region between 34.1 and 41.67 Mb on chromosome 3, and the genomic region between 12.2 and 12.7 Mb on chromosome 5 (Figure 2 and Table 4). The significant genomic region on chromosome 3 had an average SNP index value of 0.18 and 0.72 for white fruit skin-pool and light-green fruit skin-pool, respectively. Similarly, the second significant genomic region on chromosome 5 displayed an average SNP index value of 0.26 and 0.71 for white fruit skin-pool and light-green fruit skin-pool, respectively. The significant genomic region on chromosome 3 had an average Δ (SNP index) value of 0.53. In contrast, significant genomic region chromosome 5 had an average Δ (SNP index) value of 0.44 at the 95% confidence interval. These results indicated the presence of significant genomic regions conferring white immature skin color in cucumber.

### 3.3. Identification of SNPs and Candidate Genes via in Silico Analysis

To identify the SNPs and potential candidate genes associated with immature fruit skin, homozygous variants that caused the change in amino acid in deduced protein sequence of genes in the QTL regions were mined with *p* < 0.05. As a result, a total of eight homozygous variants (six SNPs and two InDels) and seven potential candidate genes were identified within the QTL region between 34.1 and 41.67 Mb on chromosome 3 (Table 5). Among the eight homozygous variants, we detected a single base pair deletion in a gene encoding LSi6 (*Cucumis sativus*) aquaporin nodulin-26-like intrinsic protein (NIP), a single base insertion within a gene encoding hypothetical protein Csa_013022 (*Cucumis sativus*) PHT [solute carrier family 15 (peptide/histidine transporter) protein], and six missense mutations (non-synonymous SNPs) in five putative genes which are known to be involved in thioredoxin-related transmembrane protein 2 isoform X1 (*Cucumis sativus*) (1 SNP); inactive poly [ADP-ribose] polymerase RCD1 (*Cucumis sativus*) (1 SNP); β-amyrin 11-oxidase (*Cucumis sativus*), AIT72036.1 cytochrome P450 (*Cucumis sativus*) (2 SNPs); Pyruvate kinase isozyme G, chloroplastic isoform X1 (*Benincasa hispida*), pyruvate kinase [EC:2.7.1.40] (1 SNP), and QWRF motif-containing protein 7 (*Cucumis sativus*) (1 SNP). By contrast, none of the homozygous variants caused the amino acid changes in the protein-coding genes between 12.2 and 12.7 Mb on chromosome 5.

### 3.4. Validation of Candidate Genes for Immature Fruit Skin Phenotype

To validate involvement of the candidate genes for immature fruit skin phenotype, a total of two CAPS and six dCAPS markers were developed based on the SNPs information available in the five putative genes (Table 6). These eight markers (M1 to M8) were firstly examined with two parental lines (MEJ and PI525075) and pooled gDNAs of F_2_ plants with light-green and white fruit skin colors, identifying that all eight markers could discriminate two skin colors in tested samples (Figure 3). Since, M1, M2, and M5 markers were based on the same SNPs information’s as M3, M4, and M6 markers, respectively. Therefore, M1, M2, M5, M7, and M8 markers were selected to be validated in 36 F_2_ plants. Marker validation assay revealed that M7 and M8 markers designed for the SNPs located at *Chr3CG52930* and *Chr3CG53640* genes were co-segregated with the light-green and white fruit skin color phenotype in 36 F_2_ plants (Figure 4). In contrast, none of the other markers (M1, M2 and M5) showed strong co-segregation among F_2_ plants (Appendix A).

These results provided strong evidence showing that the *Chr3CG52930* gene-encoding pyruvate kinase isozyme G chloroplastic isoform X1 (*Benincasa hispida*) and *Chr3CG53640* gene-encoding QWRF motif-containing protein 7 (*Cucumis sativus*) are involved in the white immature fruit skin color phenotype in PI525075. Further, BLAST search showed that *Chr3CG52930* and *Chr3CG53640* genes from JEF Korean cucumber genome have 99 and 100 % sequence homology with the *Csa3G904080* and *Csa3G915140* genes of ’Chinese Long v2’ cucumber genome, respectively.

## 4. Discussion

The immature fruit skin color is a major quality trait in cucumber, which is an important factor in determining consumer preference according to region. Likewise, chlorophyll is a primary natural pigment in the green peel of cucumber, resulting from chlorophyll biosynthesis [3,13]. A recent study suggests that white immature fruit skin color in cucumbers results from the lack of chlorophyll synthesis during fruit development [2], although several genes involved in chlorophyll biosynthesis are well known in flowering plants [22] but remain unclear in cucumber. Likewise, earlier studies have shown that white immature fruit skin color was a recessive trait and controlled mainly by a single recessive gene in cucumber [2,16]. In the present study, we have investigated the inheritance of white immature fruit skin color using an F_2_ population derived from a cross between MEJ and PI525075, indicating that a single recessive gene controls the white immature fruit skin color of cucumber. Therefore, the present study’s results are in accordance with the previous reports showing the single recessive gene inheritance for white immature fruit skin trait in cucumber [2,16].

With the growing popularity of next-generation sequencing (NGS), sequencing of the crop plants enables the detection of variants across the genome [23,24,25,26]. QTL-seq analysis allows rapid detection of the homozygous variants of a given phenotype by whole-genome resequencing of two bulked populations [18,27]. QTL-seq makes use of combined benefits of bulk-segregating analysis (BSA) and whole-genome resequencing, which can be used to identify the genomic regions responsible for the mutant phenotype in a single step [18,28,29,30]. The QTL-seq technique was recently applied to the cucumber population to identify the putative variants closely linked to causal genes responsible for subgynoecy, powdery mildew resistance, and light-green immature fruit skin traits in cucumber [13,31,32]. Although QTL-seq is a powerful tool, cucumber breeding has been largely dependent on conventional molecular mapping of putative genes, which is time consuming and requires large-scale DNA markers and many generations of advanced segregating populations [2,16,33]. Here we applied QTL-seq to detect the putative genes for the white immature fruit skin color trait in cucumber using an F_2_ population derived from a cross between the MEJ (Korean type light-green skin color) and PI525075 (white skin color).

In the past, several genes controlling the immature fruit skin color have been reported in cucumber [2,3,13,14,16]. Liu et al. mapped and identified a single base pair insertion in the *APRR2* gene, resulting in a premature stop codon, which further disrupts the accumulation of chlorophyll and chloroplast development, leading to white immature fruit skin color in cucumber [2,15]. Furthermore, two markers based on InDel (LH392580) and SNP (ASPCR39250) were developed and validated within the 8.2 kb physical interval of the *APRR2* gene [15]. Similarly, the latest study in cucumber identified the *w_0_* gene for white immature fruit color on chromosome 3 to a 100.3 kb region containing 13 candidate genes between two flanking markers, Q138 and Q193 [16]. However, this study proposed that the *Csa3G904140* gene (*w_0_*) is responsible for the white immature fruit skin color in cucumber. This study further highlighted that further study is required to validate whether the *w_0_* gene was the same as the *APRR2* gene reported in the previous study [15].

In contrast, our study identified the 1080 genes in the target genomic region based on QTL-seq analysis (Appendix A); as a result, seven candidate genes were predicted for the immature fruit skin color within the QTL region on chromosome 3. Further, homozygous variants within these seven candidates were validated via SNP marker analysis, resulting in two SNP markers (M7 and M8) developed for the missense mutations showing co-segregation with the light-green and white fruit skin-colored F_2_ plants. M7 marker designed at the nucleotide position of 6253 (P338L) of *Chr3CG52930* gene, whereas M8 marker designed at a nucleotide position of 511 (A171T) at *Chr3CG53640* gene on chromosome 3. Thus, it was likely that two genes located 0.5 Mb apart on chromosome 3 were solid putative genes for immature fruit skin color of cucumber, which laid a solid foundation for understanding the genetic basis of immature fruit skin color in cucumber.

Further, the *Chr3CG52930* gene-encoding pyruvate kinase isozyme G chloroplastic isoform X1 has been found to share 99% sequence homology with the *Csa3G904080* gene of the ‘Chinese Long v2’ cucumber genome. In the latest study, validation of three mutations in the *Csa3G904080* gene showed inconsistent results with the green and white cucumbers phenotype, which further concluded that the *Csa3G904080* gene was not a putative gene responsible for white pigmentation in cucumber [16]. Although Tang et al. [16] showed that expression of the *Csa3G904080* gene was higher in root and leaf than in fruit skin, the present study identified a novel missense mutation in the *Csa3G904080* gene and revealed a consistent result with the phenotype of green- and white-skinned cucumbers via SNP marker analysis. Therefore, these results indicated that the possible involvement of the *Csa3G904080* gene with white pigmentation requires further study due to their role in chloroplast biogenesis. Similarly, the *Chr3CG53640* gene encoding QWRF motif-containing protein 7 (*Cucumis sativus*) shows 100% sequence homology with the *Csa3G915140* gene of the ‘Chinese Long v2’ cucumber genome. A recent study has shown that a mutation in the gene-encoding QWRF motif-containing protein alters chlorophyll synthesis and reduces chlorophyll accumulation in Arabidopsis [34]. Therefore, we speculate that mutation in a *Chr3CG53640* gene could be responsible for the white pigmentation, resulting in reduced chlorophyll accumulation in immature fruit skin of the PI525075 cucumber line. Taken together, our study identified a novel allelic variant of the *Csa3G904080* gene and a new candidate gene, *Chr3CG53640*, for white immature skin color in cucumber via QTL-seq and SNP marker analyses. Hence, this study proposes a novel genetic resource controlling white immature skin color that has not been reported or validated in the previous studies. However, studying the gene function in vivo is necessary to confirm the association of *Csa3G904080* and *Chr3CG53640* genes with the white immature skin color in cucumber. Overall, this study provided a new genetic basis of immature fruit skin color trait in cucumber and is not limited to *APRR2* and *w_0_* genes, thus contributing to the development of cucumber cultivars by introgression of useful genes between the two variety groups with different skin colors. 

## Figures and Tables

**Figure 1 plants-10-02341-f001:**
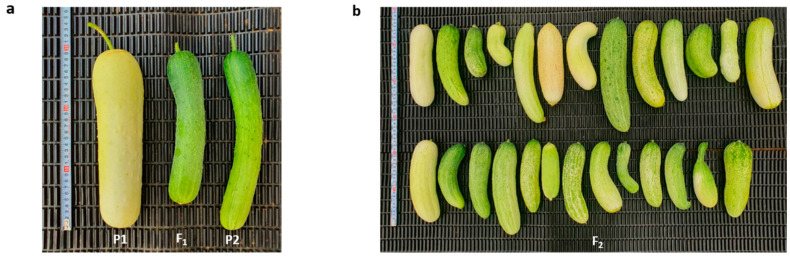
Phenotypic analysis of immature fruit skin color of immature cucumber fruits. (**a)** Fruit phenotype of PI525075 (P1), MEJ/PI525075 (F_1_) and MEJ (P2); (**b**) fruit phenotype of F_2_ derived from a cross between MEJ and PI525075.

**Figure 2 plants-10-02341-f002:**
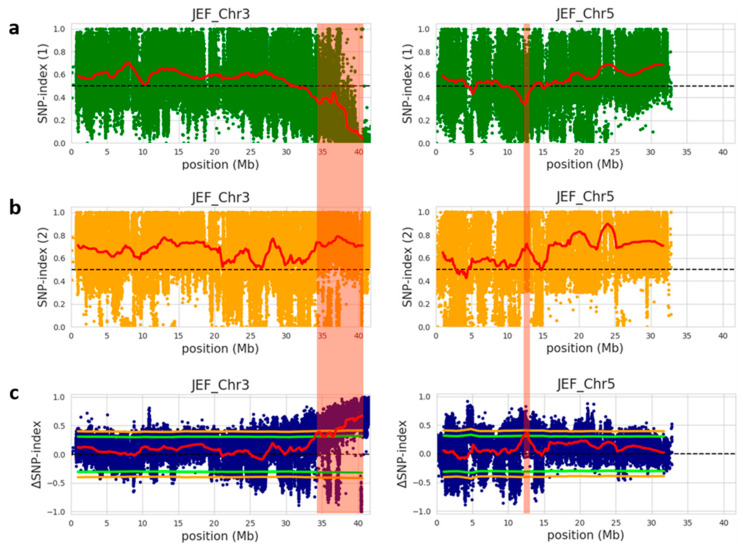
Identification of QTLs on chromosome 3 and 5 for immature fruit skin color based on QTL-seq analysis of F_2_ population. (**a**) SNP index graph of white-bulk (green dots and red line represents SNP index and sliding window average of SNP index, respectively); (**b**) SNP-index graph of light-green-bulk (orange dots and red line indicates SNP index and sliding window average of SNP index, respectively); (**c**) Δ (SNP index) graph with statistical confidence intervals under the null hypothesis of no QTLs (green line, *p* < 0.05; orange line, *p* < 0.01) from QTL-seq analysis. The significant genomic regions with *p* < 0.01 are highlighted by red shaded bar. Blue dots and red line show Δ (SNP index) and sliding window average of Δ (SNP index), respectively. SNP and Δ (SNP indexes) were calculated based on 2 Mb interval with a 100 kb sliding window.

**Figure 3 plants-10-02341-f003:**
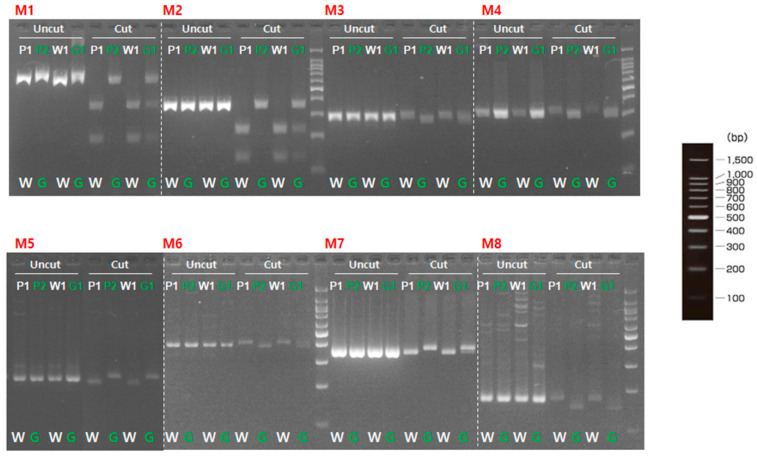
Marker validation with parental lines and bulked F_2_ plants. P1, ‘PI525075’ (white); P2, ‘MEJ’ (light-green); W1, white pool, G1, light-green pool; W, white; G, light-green.

**Figure 4 plants-10-02341-f004:**
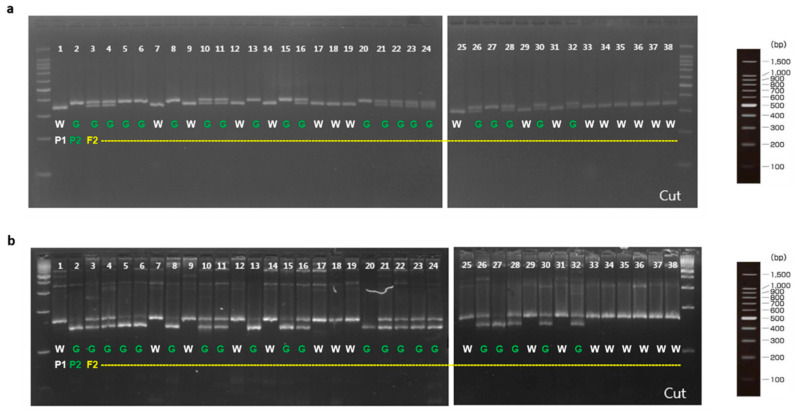
dCAPS analysis of novel alleles in the F_2_ plants derived from a cross between MEJ and PI525075. P1, ‘PI525075’ (white); P2, ‘MEJ’ (light-green); W, white; G, light-green. (**a**) M7 marker for a novel allele of *Chr3CG52930*; (**b**) M8 marker for a novel of allele of *Chr3CG53640*.

**Table 1 plants-10-02341-t001:** Segregation analysis of immature fruit skin color phenotype in cucumber.

Name	Population	Plant Number	Immature Fruit Skin Color Phenotype				
			White	Light-Green	Expected	χ^2^	*p* Value ^†^
PI525075	P1	10	10	0	-	-	-
MEJ	P2	10	0	10	-	-	-
MEJ/PI525075	F_1_	10	0	10	-	-	-
MEJ/PI525075	F_2_	136	30	106	3:1	0.62	0.42

^†^ Not significant (*p* Value > 0.05).

**Table 2 plants-10-02341-t002:** Summary of whole genome re-sequencing data used for QTL-seq analysis.

Samples	Raw Data		Trimmed Data			Coverage (X)
	Reads	Read Length (bp)	Reads	Read Length (bp)	%	
PI525075	192,314,600	29,039,504,600	157,678,304	22,512,072,535	77.52	64.32
MEJ	191,719,994	28,949,719,094	187,218,180	27,266,475,876	94.19	77.90
White-pool	204,951,660	30,947,700,660	175,957,140	25,010,424,993	80.82	71.46
Light-green-pool	228,647,910	34,525,834,410	197,713,150	28,154,853,131	81.55	80.44

**Table 3 plants-10-02341-t003:** Polymorphisms identified in QTL-seq analysis.

Chromosome Number	No. of SNPs	
	White-Pool	Light-Green-Pool
1	11,896	8263
2	21,047	20,044
3	46,480	16,303
4	17,843	25,400
5	27,535	15,122
6	18,782	21,018
7	16,809	14,749
Total	160,392	120,899

**Table 4 plants-10-02341-t004:** Candidate QTLs associated with immature fruit skin color of cucumber based on QTL-seq analysis.

Chr.	Physical Position (Mb)	Size (bp)	SNPs	Genes
3	34.1–41.67	7,579,492	11,435	556
5	12.2–12.7	500,000	743	33

**Table 5 plants-10-02341-t005:** Identification of sequence variations and candidate genes in the QTL region on chromosome 3.

Position	Ref	Alt	Effect	AA	Type	Gene ID	Description
34626707	TAAAAAAAAAAA	TAAAAAAAAAA	Deletion	ATG	InDel	*Chr3CG43770*	LSi6 [*Cucumis sativus* var. *sativus*], aquaporin NIP
40443941	T	C	Non_synonymous	I3V	SNP	*Chr3CG51850*	Thioredoxin-related transmembrane protein 2 isoform X1 [*Cucumis sativus*]
40660199	T	G	Non_synonymous	L457R	SNP	*Chr3CG52290*	Inactive poly [ADP-ribose] polymerase RCD1 [*Cucumis sativus*]
41075124	G	T	Non_synonymous	V250L	SNP	*Chr3CG52880*	β-amyrin 11-oxidase [*Cucumis sativus*], AIT72036.1 cytochrome P450 [*Cucumis sativus*]
41075140	T	G	Non_synonymous	I255R	SNP	*Chr3CG52880*	β-amyrin 11-oxidase [*Cucumis sativus*], AIT72036.1 cytochrome P450 [*Cucumis sativus*]
41102542	CTTTTTTTTTTTTT	CTTTTTTTTTTTTTT	Insertion	S123F	InDel	*Chr3CG52910*	Hypothetical protein Csa_013022 [*Cucumis sativus*], PHT; solute carrier family 15 (peptide/histidine transporter)
41106529	G	A	Non_synonymous	P338L	SNP	*Chr3CG52930*	Pyruvate kinase isozyme G, chloroplastic isoform X1 [*Benincasa hispida*], pyk; pyruvate kinase [EC:2.7.1.40]
41641571	G	A	Non_synonymous	A171T	SNP	*Chr3CG53640*	QWRF motif-containing protein 7 [*Cucumis sativus*]

**Table 6 plants-10-02341-t006:** Primers used for genotyping the parents and F_2_ plants.

Marker Name	Chr	Position	Gene ID	Marker Type	Forward/Reverse Sequence (5′–3′)	Restriction Site	Restriction Enzyme	Amplicon Size (Light-Green vs. White)
M1	3	40443941	*Chr3CG51850*	CAPS	TTCTCTAATGACGATGACGTTG/ATCCGGAATTTCTTTCTTCTTC	ACGT	*Hpy*CH4IV	691 vs. (264, 427)
M2	3	40660199	*Chr3CG52290*	CAPS	TCTGTGAACTTTGACAGTGGAG/TAGATCTCCACAAGTCTCCCAT	ACGT	*Hpy*CH4IV	360 vs. (226, 134)
M3	3	40443941	*Chr3CG51850*	dCAPS	TTTCCGGGATGAATTCCCGGATCGA/ATCCGGAATTTCTTTCTTCTTC	ATCGAT	*Cla*I	238 vs. 263
M4	3	40660199	*Chr3CG52290*	dCAPS	AAGATACCATCAAGTCCTCTTAAGC/GCAATTTATCTGCAACTGGTCT	AAGCTT	*Hind*III	272 vs. 297
M5	3	41075124	*Chr3CG52880*	dCAPS	GAAGATAGAGAATACAATTCAAGCT/TATGTAGAAGAGCCCAACAAGC	AAGCTT	*Hind*III	197 vs. 172
M6	3	41075140	*Chr3CG52880*	dCAPS	TGTTTTATCATTTTCAAATCTACGT/GCCTATTAAATTGGTGGATGAA	TACGTA	*Sna*BI	338 vs. 363
M7	3	41106529	*Chr3CG52930*	dCAPS	AACCTTGTGGACACTCGATGGACTT/ATGCGTGTTCCTCTAGTTTGTT	CTNAG	*Dde*I	327 vs. 302
M8	3	41641571	*Chr3CG53640*	dCAPS	ATGGAAGTCTCTGCTCTAACCTAAG/AAACTAGGCAGTCAACGAGGT	CCTNAGC	*Bpu*10I	113 vs. 138

## Data Availability

All relevant raw sequence data generated in this study are available in the Sequence Read Archive (SRA) of the NCBI database with the following accession number: MEJ (SRX10261885) and PI525075 (SRR16093564).

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
