# Peer review of "Development of SNP Markers for White Immature Fruit Skin Color in Cucumber (*Cucumis sativus* L.) Using QTL-seq and Marker Analyses"

_plants, 2021, doi:10.3390/plants10112341_

Round 1

Reviewer 1 Report

Review manuscript „Development of SNP markers for white immature fruit skin color in cucumber (Cucumis sativus L.) using QTL-seq analysis“

Authors submitted a highly interesting manuscript on the genetic basis of the trait white immature fruit skin color in cucumber. They have shown a monogenic recessive inheritance of the gene and identified two QTLs on chromosome 3 and 5 of cucumber using QTL-seq analysis combined with bulked segregant analysis. dCAPS markers were validated in a subset of the segregating F2-progeny and candidate genes were discussed. The manuscript is clearly structured and written well.

Nethertheless there are some open questions:

  • Table 3: The greatest difference in number of SNPs between White-pool and Light-green-pool is found on chromosome 3. Can this be an effect of target gene location on this chromosome or is this by chance?
  • Line 195/196: You described (a small and not very high) QTL on chromosome 5, but you negate this QTL in the complete further manuscript. Have you tested markers from the segment 12.2-12.7 as you have done it for the markers on chromosome 3? Here you must write at least one sentence.
  • Figure 2: For chromosome 3 in a), b) and c) the significance intervall as well as the lines end before the chromosome end. Is this a methodical artefact by sliding window average? Or how you can explain this? This region is especially interesting because the Δ SNP index is still increasing.
  • Line 240: For Fig. 4 it is completely enougth to show the results for 36 F2-plants (one gel). But you have not analysed the complete progeny of 136 plants? Have you found in the complete progeny recombinations between dCAPS markers and the target gene (phenotype) this would give important informations on genetic distance. Please add these data at least in the text.
  • Line 253 ff + discussion: You have shown monogenic recessive inheritance of white immature fruit skin color and now you discuss the involvement of two genes, pyruvate kinase isozyme G chloroplastic isoform X1 and QWRF motif-containing protein 7, for expression of the phenotype. It should be clear, that only one gene is involved, otherwise your genetic of the trait is wrong. So it can be the first gene or the second or a complete other, because the region 34.1-41.67 is very large and you have no final evidence for the causing gene. This can be clarified only by complementation or knock out strategy or at least an comparative expression analysis. But this is in the moment not necessary for usage oft he markers in breeding research and breeding.

Author Response

For reviewer #1:

Authors submitted a highly interesting manuscript on the genetic basis of the trait white immature fruit skin color in cucumber. They have shown a monogenic recessive inheritance of the gene and identified two QTLs on chromosome 3 and 5 of cucumber using QTL-seq analysis combined with bulked segregant analysis. dCAPS markers were validated in a subset of the segregating F2-progeny and candidate genes were discussed. The manuscript is clearly structured and written well.

  1. Table 3: The greatest difference in number of SNPs between White-pool and Light-green-pool is found on chromosome 3. Can this be an effect of target gene location on this chromosome or is this by chance?

Ans: Thank you for your comment and suggestion. We carefully performed phenotyping and selected the plants for obtaining genomic DNAs from the White-pool and Light-green-pool. Therefore, we think that the greatest difference of SNP number between both pools on chromosome 3 are resulted from target QTL harboring target gene. We also think that this was well confirmed by molecular marker derived from variants within the QTL in this study.

  1. Line 195/196: You described (a small and not very high) QTL on chromosome 5, but you negate this QTL in the complete further manuscript. Have you tested markers from the segment 12.2-12.7 as you have done it for the markers on chromosome 3? Here you must write at least one sentence.

Ans: Thank you for your comment and suggestion. The manuscript has now been thoroughly revised considering your suggestions. When we applied selection criteria such as (1) Variants showing differences between bulk1 (white skin) and bulk2 (light-green skin) were selected. (2) Among the selected variants, only homozygous variants were selected. (3) Homozygous variants causing the amino acid change in protein sequences encoded by genes, we could not find any gene or variant within QTL on chromosome 5. As your comment, we added the sentence “By contrast, none of the homozygous variants caused the amino acid changes in the protein-coding genes between 12.2 and 12.7 Mb on chromosome 5” at Lines 203-204.

  1. Figure 2: For chromosome 3 in a), b) and c) the significance interval as well as the lines end before the chromosome end. Is this a methodical artefact by sliding window average? Or how you can explain this? This region is especially interesting because the Δ SNP index is still increasing.

Ans: Thank you for your comment and suggestion. We think that the interval at the end of chromosome 3 in SNP-index plot was caused by methodical limitation of QTL-seq program that uses the central position of sliding window to draw SNP-index plot. Considering the limitation, we already extended QTL region on chromosome 3 to the end (41679492) of chromosome 3 and used for further study.

  1. Line 240: For Fig. 4 it is completely enough to show the results for 36 F2-plants (one gel). But you have not analyzed the complete progeny of 136 plants? Have you found in the complete progeny recombination between dCAPS markers and the target gene (phenotype) this would give important information’s on genetic distance? Please add these data at least in the text.

Ans: Thank you for your comment and suggestion. We did not apply the dCAPS markers to 136 F2 plants because of cost and time. And, while implementing the experiment, we judged that the size of sample was enough to validate the marker practically. Among the five dCAPS markers (M1, M2, M5, M7 and M8), we found recombination between the markers and phenotype in the three markers (M1, M2 and M5) except other two markers (M7 and M8). Thus, we thought that the two markers (M7 and M8) were more closely associated with phenotype than the other three markers (M1, M2 and M5). Considering your comment, advanced research would be needed to determine how much recombination occur between the two closely-linked markers in a bigger segregating population and to conclude major gene affecting the skin color in the future study. By considering your suggestion, we have added Table S1 to show the total number genes (1080) and their protein description within the QTL region of chromosome 3.  

  1. Line 253 ff + discussion: You have shown monogenicrecessive inheritance of white immature fruit skin color and now you discuss the involvement of two genes, pyruvate kinase isozyme G chloroplastic isoform X1 and QWRF motif-containing protein 7, for expression of the phenotype. It should be clear, that only one gene is involved, otherwise your genetic of the trait is wrong. So it can be the first gene or the second or a complete other, because the region 34.1-41.67 is very large and you have no final evidence for the causing gene. This can be clarified only by complementation or knock out strategy or at least a comparative expression analysis. But this is in the moment not necessary for usage of the markers in breeding research and breeding.

Ans: Thank you for your valuable comment and suggestion. This study was conducted to understand the genetic basis of white immature fruit skin color in cucumber and to develop a molecular marker closely linked to the gene. However, we found two putative genes via QTL-seq and SNP marker analyses, thus contributing to the cloning of putative genes and development of cucumber cultivars using marker-assisted breeding. In the current study, we could not be able to propose a main controlling gene for the phenotype but suggested two candidate genes for white skin color, one of which is a new candidate gene in cucumber.  By considering your suggestion, we have added Table S1 to show the total number genes (1,080) and their protein description within the QTL region of chromosome 3.  

  1. Previously Tang et al. 2018 showed that Csa3G904080 gene encoding pyruvate kinase isozyme G chloroplastic isoform X1expression was elevated in root and leaf than in fruit skin but the present study identified a novel allelic variant of the Csa3G904080 gene and revealed a consistent result with the phenotype of green and white skin cucumbers. However, we think that possible involvement of Csa3G904080 gene with the white pigmentation can’t be ignored due to their role in the chloroplast biogenesis.  

Reference: Tang, H.-Y.; Dong, X.; Wang, J.-K.; Xia, J.-H.; Xie, F.; Zhang, Y.; Yao, X.; Xu, Y.-J.; Wang, Z.-J. Fine Mapping and Candidate Gene Prediction for White Immature Fruit Skin in Cucumber (Cucumis sativus L.). Int J Mol Sci 201819, 1493, doi:10.3390/ijms19051493.

  1. Similarly, studies (Albrecht et al 2010) in Arabidopsis is evident that gene encoding QWRF motif-containing protein alters chlorophyll synthesis and reduces chlorophyll accumulation. Therefore, we speculate that mutation in a Chr3CG53640 gene could be responsible for the white pigmentation, resulting in reduced chlorophyll accumulation in immature fruit skin of the PI525075 cucumber accession.

Reference: Albrecht, V.; Šimková, K.; Carrie, C.; Delannoy, E.; Giraud, E.; Whelan, J.; Small, I.D.; Apel, K.; Badger, M.R.; Pogson, B.J. The Cytoskeleton and the Peroxisomal-Targeted SNOWY COTYLEDON3 Protein Are Required for Chloroplast Development in Arabidopsis  The Plant Cell 2010, 22, 3423-3438, doi:10.1105/tpc.110.074781.

However, as you commented, studying the gene expression and gene function in vivo is necessary to confirm the association of Csa3G904080 and Chr3CG53640 genes with the white immature skin color in cucumber.  The current experimental design did not allow us to perform complementation or knock out strategy or at least a comparative expression analysis. Therefore, we are working on cloning and expression analysis of the putative genes at different fruit developmental stages in our future study.    

Reviewer 2 Report

Dear Authors, 
When reviewing scientific papers for publication, I usually start with a general overview in terms of structure,the abstract, literature review, methodology, the research findings, discussion, conclusions, as well as limitations of the study.
Please find my comments below.
1.    What gaps in knowledge would you like to address as the purpose of this paper? Presentation of technical research results may not be suitable to fill knowledge gaps in the field of your study. Please clearly address major challenges regarding the research topic and your novel contribution(s) to your field and in the context of your study.
2.    I couldn't identify any hypotheses emphasized in the text.
3.    I recommend adding some references to the latest subject literature (including Web of Science and Scopus papers).
4.    I also believe the catalog of the literature cited is rather poor. I suggest expanding the list of literature studies by the years 2020-2021
5.    I am not fully qualified to evaluate your English, but it appears that a professional native speaker must be involved to improve your writing.

Author Response

For reviewer #2:

When reviewing scientific papers for publication, I usually start with a general overview in terms of structure, the abstract, literature review, methodology, the research findings, discussion, conclusions, as well as limitations of the study.

  1. What gaps in knowledge would you like to address as the purpose of this paper? Presentation of technical research results may not be suitable to fill knowledge gaps in the field of your study. Please clearly address major challenges regarding the research topic and your novel contribution(s) to your field and in the context of your study.

Ans: Thank you for your valuable comment and suggestion. The manuscript has now been thoroughly revised considering your suggestions. This study was conducted to understand the genetic basis of white immature fruit skin color in cucumber and develop a molecular maker tightly linked to the phenotype applicable for breeding of Korean type of slicer cucumber which have two-toned skin color of light green back ground and dark green stalk-end color. However, previous reports described genetic basis on uniform dark green skin color. The present study identified a novel missense mutation in the Csa3G904080 gene and a new candidate gene, Chr3CG53640, for white immature skin color in cucumber via QTL-seq and SNP marker analyses. Hence, this study contributing to the cloning of putative genes and development of cucumber cultivars using marker-assisted breeding.   

  1. I couldn't identify any hypotheses emphasized in the text.

Ans: Thank you for your comment and suggestion. The manuscript has now been thoroughly revised considering your suggestions.  The present study identified a novel missense mutation in the Csa3G904080 gene and a new candidate gene, Chr3CG53640, for white immature skin color in cucumber via QTL-seq and SNP marker analyses. Hence, this study contributing to the cloning of putative genes and development of cucumber cultivars using marker-assisted breeding.   

  1. I recommend adding some references to the latest subject literature (including Web of Science and Scopus papers).

Ans: Thank you for your comment and suggestion. The manuscript has now been thoroughly revised considering your suggestions.  We added 10 additional relevant references from Web of Science and Scopus.

Varshney, R.K.; Nayak, S.N.; May, G.D.; Jackson, S.A. Next-generation sequencing technologies and their implications for crop genetics and breeding. Trends in Biotechnology 2009, 27, 522-530, doi:https://doi.org/10.1016/j.tibtech.2009.05.006.

Abe, A.; Kosugi, S.; Yoshida, K.; Natsume, S.; Takagi, H.; Kanzaki, H.; Matsumura, H.; Yoshida, K.; Mitsuoka, C.; Tamiru, M.; et al. Genome sequencing reveals agronomically important loci in rice using MutMap. Nature Biotechnology 2012, 30, 174-178, doi:10.1038/nbt.2095.

Sui, X.; Shan, N.; Hu, L.; Zhang, C.; Yu, C.; Ren, H.; Turgeon, R.; Zhang, Z. The complex character of photosynthesis in cucumber fruit. Journal of Experimental Botany 2017, 68, 1625-1637, doi:10.1093/jxb/erx034.

Win, K.T.; Zhang, C.; Silva, R.R.; Lee, J.H.; Kim, Y.-C.; Lee, S. Identification of quantitative trait loci governing subgynoecy in cucumber. Theoretical and Applied Genetics 2019, 132, 1505-1521, doi:10.1007/s00122-019-03295-3.

D S, K.; Chung, S.-M. Development of SNP markers and validation assays in commercial Korean melon cultivars, using Genotyping-by-sequencing and Fluidigm analyses. Scientia Horticulturae 2019, 263.

Kishor, D.S.; Seo, J.; Chin, J.H.; Koh, H.-J. Evaluation of Whole-Genome Sequence, Genetic Diversity, and Agronomic Traits of Basmati Rice (Oryza sativa L.). Frontiers in Genetics 2020, 11, doi:10.3389/fgene.2020.00086.

Ramos, A.; Fu, Y.; Michael, V.; Meru, G. QTL-seq for identification of loci associated with resistance to Phytophthora crown rot in squash. Scientific Reports 2020, 10, 5326, doi:10.1038/s41598-020-62228-z.

Wang, M.; Chen, L.; Liang, Z.; He, X.; Liu, W.; Jiang, B.; Yan, J.; Sun, P.; Cao, Z.; Peng, Q.; et al. Metabolome and transcriptome analyses reveal chlorophyll and anthocyanin metabolism pathway associated with cucumber fruit skin color. BMC Plant Biol 2020, 20, 386, doi:10.1186/s12870-020-02597-9.

Kishor, D.S.; Noh, Y.; Song, W.-H.; Lee, G.P.; Park, Y.; Jung, J.-K.; Shim, E.-J.; Sim, S.-C.; Chung, S.-M. SNP marker assay and candidate gene identification for sex expression via genotyping-by-sequencing-based genome-wide associations (GWAS) analyses in Oriental melon (Cucumis melo L.var.makuwa). Scientia Horticulturae 2021, 276, 109711, doi:https://doi.org/10.1016/j.scienta.2020.109711.

Yang, L.; Lei, L.; Li, P.; Wang, J.; Wang, C.; Yang, F.; Chen, J.; Liu, H.; Zheng, H.; Xin, W.; et al. Identification of Candidate Genes Conferring Cold Tolerance to Rice (Oryza sativa L.) at the Bud-Bursting Stage Using Bulk Segregant Analysis Sequencing and Linkage Mapping. Front Plant Sci 2021, 12, doi:10.3389/fpls.2021.647239.

  1. I also believe the catalog of the literature cited is rather poor. I suggest expanding the list of literature studies by the years 2020-2021

Ans: Thank you for your comment and suggestion. The manuscript has now been thoroughly revised considering your suggestions. We have added latest references by the years 2020-21.

Kishor, D.S.; Seo, J.; Chin, J.H.; Koh, H.-J. Evaluation of Whole-Genome Sequence, Genetic Diversity, and Agronomic Traits of Basmati Rice (Oryza sativa L.). Frontiers in Genetics 2020, 11, doi:10.3389/fgene.2020.00086.

Ramos, A.; Fu, Y.; Michael, V.; Meru, G. QTL-seq for identification of loci associated with resistance to Phytophthora crown rot in squash. Scientific Reports 2020, 10, 5326, doi:10.1038/s41598-020-62228-z.

Wang, M.; Chen, L.; Liang, Z.; He, X.; Liu, W.; Jiang, B.; Yan, J.; Sun, P.; Cao, Z.; Peng, Q.; et al. Metabolome and transcriptome analyses reveal chlorophyll and anthocyanin metabolism pathway associated with cucumber fruit skin color. BMC Plant Biol 2020, 20, 386, doi:10.1186/s12870-020-02597-9.

Kishor, D.S.; Noh, Y.; Song, W.-H.; Lee, G.P.; Park, Y.; Jung, J.-K.; Shim, E.-J.; Sim, S.-C.; Chung, S.-M. SNP marker assay and candidate gene identification for sex expression via genotyping-by-sequencing-based genome-wide associations (GWAS) analyses in Oriental melon (Cucumis melo L.var.makuwa). Scientia Horticulturae 2021, 276, 109711, doi:https://doi.org/10.1016/j.scienta.2020.109711.

Yang, L.; Lei, L.; Li, P.; Wang, J.; Wang, C.; Yang, F.; Chen, J.; Liu, H.; Zheng, H.; Xin, W.; et al. Identification of Candidate Genes Conferring Cold Tolerance to Rice (Oryza sativa L.) at the Bud-Bursting Stage Using Bulk Segregant Analysis Sequencing and Linkage Mapping. Front Plant Sci 2021, 12, doi:10.3389/fpls.2021.647239.

  1. I am not fully qualified to evaluate your English, but it appears that a professional native speaker must be involved to improve your writing.

Ans: Thank you for your comment and suggestion. Revised as per the reviewer suggestion. We have tried to improve our English writing through revision from professional native speaker at Sejong University. We also verified the grammar and sentence structure via Grammarly. 

Round 2

Reviewer 2 Report

Dear Authors,
thanks for the implemented corrections, which significantly improved the quality of the manuscript.

I wish you success in your research work.